# Effect of Pro-Ecological Cooling and Lubrication Methods on the Sharpening Process of Planar Blades Used in Food Processing

**DOI:** 10.3390/ma15217842

**Published:** 2022-11-07

**Authors:** Bartosz Zieliński, Krzysztof Nadolny, Wojciech Zawadka, Tomasz Chaciński, Wojciech Stachurski, Gilmar Ferreira Batalha

**Affiliations:** 1Espersen Koszalin Sp. o.o., Mieszka I 29, 75-124 Koszalin, Poland; 2Department of Production Engineering, Faculty of Mechanical Engineering, Koszalin University of Technology, Racławicka 15-17, 75-620 Koszalin, Poland; 3Institute of Machine Tools and Production Engineering, Lodz University of Technology, Stefanowskiego 1/15, 90-537 Lodz, Poland; 4Escola Politecnica da Universidade de Sao Paulo, São Paulo 01246-904, SP, Brazil

**Keywords:** grinding process, pro-ecological cooling methods, cutting blade, food processing

## Abstract

This work presents the results of an experimental study of the sharpening of planar technical blades used in the fish processing industry. Sharpening was carried out in the grinding process using several environmentally friendly methods of cooling and lubricating the machining zone (MQL method, CAG nozzle, hybrid method that is a combination of MQL and CAG methods, as well as WET flooding method as reference). The purpose of the research was to determine the possibility of reducing the negative environmental impact of the sharpening process of technical blades by minimizing the expenditure of coolant. The application of the MQL method and the hybrid MQL + CAG method provided a very good realization of the lubricating function so that the share of friction of dulled cutting vertices against the workpiece surface is reduced, which manifests itself in the reduction of the grinding force and the correlated grinding power. In the case of grinding under cooled compressed air delivery conditions, the average cutting force was as much as 91.6% higher (*F* = 22.63 N) compared to the result obtained for the most favorable flooding method, demonstrating the insufficient quality of the blade shaped under such conditions. A comprehensive comparison of test results on grinding power gain, cutting force and surface texture suggests that the most favorable sharpening results were obtained using the environmentally friendly MQL method of cooling and lubricating the grinding zone.

## 1. Introduction

A technical blade is called a tool used to separate in the process of cutting, cutting or notching non-metallic materials. Many tools designed for cutting non-metallic materials are produced in the world and, due to their prevalence, are used in many industries, such as meat processing, fish processing, vegetable and fruit processing, zoology, general surgery, forensic medicine, and others. The correctness of the blade’s work has a major impact on both the material being cut through, but also on the person using the tool or machine [1].

Despite the significant number of processes carried out with technical blades, research is still being conducted to determine the so-called blade sharpness index (BSI). Some of the authors of the publications state that the BSI depends on the required force to be applied to separate a known material, while others make it dependent on the measurement of the angle and radius of the blade edge [2,3].

Operation of insufficiently sharp tools leads to many musculoskeletal disorders as a result of excessive force used by workers [4]. In the case of machinery, a worn tool causes excessive energy consumption and also premature wear of operating parts affected by excessive force [5].

The characteristics of cutting tools are determined by many factors. They can be divided by their use as either manual or machine and subsequently by their shape. In the case of machine use, blades can take the shape of rectilinear, disc, cylindrical, or belt (straight and serrated), shaped. Technical blades can be shaped as one, two and even three-piece. They occur as symmetrical and asymmetrical. The durability of the blade and cutting forces depend directly on the value of the blade vertex angle *α*, and the values that are commonly used in the food industry are in the range of *α* = 3–10° [6]. The limitation of the minimum value of this angle is due to the durability of the blade and the accompanying phenomenon of blade edge rewinding at the machining stage [1,7].

Due to the wide collection of types of technical blades, this paper focuses on technical blades applicable to the cutting of organic materials. The leading material for technical blades in the food industry is stainless steel. Ceramic materials are another increasingly common group of materials. According to some sources, the first ceramic knives appeared as early as the 1980s in Japan [8].

Technical blades made of stainless steel are mainly shaped in the grinding process, and depending on their intended use, they can be subjected to finishing machining in the form of lapping, polishing or electrochemical polishing, among others [8]. This process, due to its dependence on many factors, makes the quality of the edge of the blade variable and achieving reproducible sharpness of the blade edges is still a challenge. The choice of machining method and tools used to obtain the final desired characteristics of the blade is very important. A comparison of the above methods is the subject of the work [6]. However, it should be noted that cutting tool manufacturing companies establish their own technological processes based on their experience and new manufacturing technologies. Tracing the available materials, one can find in the patent offices alternative methods of manufacturing technical blades from stainless steel. Despite this, the most widely used method of shaping technical blades is still abrasive machining [8].

Grinding processes are characterized by a high proportion of friction due to the undefined geometry of the abrasive grains and their random orientation on the active surface of the abrasive tools, which results in the generation of a large amount of heat and forces the use of cooling and lubricating agents for the efficient implementation of this process [9,10].

The use of coolants, lubricants and anti-adhesives in grinding processes involves high costs, and, in addition, many of the coolants and lubricants commonly used negatively affect the environment and human health. Therefore, for many years there has been an intensification of the development of methods to minimize their expenditure. An alternative to the use of lubricating liquids are methods of gas supply using a cold air gun (CAG) or cryogenic cooling method (liquid CO_2_ or N_2_), which provide a realization of the cooling function but insufficiently lubricate the grinding zone [11,12,13,14,15,16,17,18].

In addition to the type, the effectiveness of the cooling-lubricating medium in the grinding process is decisively influenced by its expenditure and the effectiveness of reaching the contact zone of the active surface of the grinding wheel with the machined surface directly. For many years, intensive research work has been carried out on the development of new methods for the supply of coolants, because of which the following methods have been developed:minimum quantity lubrication MQL [19,20,21,22];minimum quantity cooling MQC [23,24];minimum quantity cooling lubrication MQCL [25,26,27,28];cooled air minimum quantity lubrication CAMQL [29,30];cold air and oil mist CAOM [31,32,33,34].

It should also be noted that it is possible to replace liquids and gases with a cooling and lubricating effect with solid substances delivered to the grinding zone, e.g., in the form of an impregnant filling the free intergranular spaces in grinding wheels. The most described substances with a lubricating and anti-adhesive effect in the solid state used in impregnated grinding wheels include sulfur, graphite, waxes, resins, hBN, MoS_2_ and silicone [17,35,36].

This work presents the results of an experimental study of the sharpening of planar technical blades used in the fish processing industry. Sharpening was carried out in the grinding process using several environmentally friendly methods of cooling and lubricating the machining zone (MQL method, CAG nozzle, hybrid method that is a combination of MQL and CAG methods, as well as WET flooding method as reference). The purpose of the research was to determine the possibility of reducing the negative environmental impact of the sharpening process of technical blades by minimizing the expenditure of coolant. In other words, the purpose of the study was to determine whether it is possible to replace the traditional flooding method with one of the pro-ecological methods of cooling and lubricating the machining zone (characterized by minimizing the expenditure of cooling-lubrication agents). This is the first comprehensive study of its kind on the abrasive processing of technical blades.

## 2. Materials and Methods

The primary objective of the experimental study was to determine what effect the cooling and lubrication conditions of the machining zone (the WET flooding method, the MQL method, the CAG nozzle, and the hybrid method, which is a combination of the MQL and CAG methods) have on the grinding process, the quality of the shaped blade and the cutting force using it.

In the study of the process of rectilinear grinding of planar knife blades intended for the process of skinning flat fish, a test stand equipped with a specialized five-axis computerized numerical control (CNC) grinder for shaping knife blades with low stiffness was used (Figure 1a), described in more detail in work [37].

Figure 2 shows a view of the grinding zone in a configuration that allows four methods of delivering coolants to the grinding zone. The tests were carried out in the process of rectilinear grinding of flat surfaces using a grinding wheel with grains of cubic boron nitride of technical designation 5A1 35 × 25 × 10/22 × 15 B181 V240 SV (Table 1). Grinding was carried out with the face of the grinding wheel, on which a conical chamfer was shaped to allow machining to be carried out with the following values of spindle axis inclination *α_s_* = 85°, *β_s_* = 5°, *χ_s_* = 20° (resulting from the preliminary studies)—Figure 3.

In the study, the value of the longitudinal feed speed of the grinding wheel was assumed to be *v_f_* = 100 mm/min. Each time a given cutting edge was first roughly ground with an allowance of *a_e rough_* = 0.10 mm, followed by a sparking-out pass with an allowance value of: *a_e_*
_*sparking-out*_ = 0.02 mm.

During the study, different methods of delivering cooling and lubrication agents were used, and their type and expenditure varied. The flooding (WET) method was used as the reference method. In addition, tests were conducted under the conditions of using the minimum quantity lubrication (MQL) method, cooled compressed air (CAG), and the conditions of using the MQL and CAG nozzle simultaneously (MQL + CAG). The use of the MQL method allows effective lubrication of the grinding zone but, in many cases, does not provide adequate heat dissipation due to the limited contribution of the lubricant. On the other hand, the use of cooled compressed air generated by the CAG nozzle allows cooling of the machining zone with a marginal realization of the lubricating function. Thus, it seems natural to combine these two methods (in one hybrid MQL + CAG method), whose features are complementary.

Prior to the grinding test, the grinding wheel was dressed each time to obtain repeatability of the machining results and eliminate the effects of wheel wear. After the planar blade is finished shaping in the grinding process, a so-called rewind of the material is created, which must be removed (usually manually) using whetstones, wood, or leather. In the described research, a special device made to achieve repeatability of this procedure was used to remove the rewind after the grinding operation (Figure 1b). The rewind was removed using leather set on a flat wooden support, which was moved several times along the machined edge of the blade in a rectilinear feed motion.

During the tests, the increase of grinding power *ΔP* (representing the difference between the maximum power during grinding and the spindle power at idle speed) was recorded (using software controlling the operation of the grinding machine), and after the tests, the values of selected surface texture parameters of the blades were determined (on a position for contact measurements of surface texture (Figure 1c), the morphology of the blades was analyzed (using a digital measuring microscope—Figure 1d) and knife cutting forces were determined after grinding on a special test stand (Figure 1e). A detailed description of the test stand, the methodology for conducting cutting tests, and the verification of its correctness are described in this work [43]. Table 2 provides a synthetic summary of the parameters and conditions with which the experimental studies were conducted.

The study used an experimental planning methodology, which resulted in a research plan and the number of repetitions of each point in the plan *n* = 5.

In the study, planar technical knives with dimensions of 459.5 mm × 12.3 mm × 0.6 mm (Kuno Wasser GmbH, Solingen, Germany, for Steen F.P.M. International, Kalmthout, Belgium) made of high-carbon martensitic stainless steel X39Cr13 were ground [37].

## 3. Results and Discussion

Figure 4 presents a chart showing the average values of the grinding power gain *ΔP* determined during grinding tests carried out with 5 repetitions using four methods of delivering coolants to the machining zone (WET, MQL, CAG and MQL + CAG). Meanwhile, Figure 5 shows the average values from the results of cutting force *F* measurements carried out with the knives shaped by the described studies.

As can be seen from the data in Figure 4, the smallest values of the grinding power gain *ΔP* were measured when the grinding process was carried out under lubrication conditions with MQL (*ΔP* = 15.20 W) and with the hybrid method MQL + CAG (*ΔP* = 14.40 W). For the other two methods of cooling and lubricating the machining zone (WET and CAG), the *ΔP* values were about 50–60% higher, amounting to *ΔP* = 22.67 W and *ΔP* = 23.20 W, respectively. The use of the MQL method alone and in combination with the CAG nozzle provided a very good realization of the lubricating function (in this method, an air-oil aerosol is supplied). Under conditions of good lubrication, the proportion of friction of dulled cutting vertices against the workpiece surface is reduced, which manifests itself as a reduction in grinding force and the correlated power, measured in the described studies as the electrical power of the grinding wheel spindle.

Comparing the average values of the cutting force *F*, summarized in the chart in Figure 5, the most beneficial cutting properties (manifested by relatively small *F* values) were obtained using flooding cooling WET (*F* = 11.81 N), MQL (*F* = 15.25 N) and the hybrid method MQL + CAG (*F* = 13.69 N) in the grinding process. In the case of grinding under cooled compressed air delivery, the average cutting force was as much as 91.6% higher (*F* = 22.63 N) compared to the result obtained for the most advantageous flooding method. In addition, it should be noted that direct observations of the cut surface carried out during the trials indicated by far the worst quality of polyurethane samples after cutting with ground blades under conditions of using only the CAG nozzle. The obtained force measurement results give rise to the conclusion of insufficient quality of the blade shaped under such conditions, which was the subject of further surface texture analysis and microscopic observations.

In the next stage of analysis, the microtopography of the surfaces of the knives shaped in the study (three blades for each method of cooling and lubricating the grinding zone) was measured. Figure 6 shows a set of sample axonometric views of the surface microtopography (with dimensions of 1.0 mm × 1.0 mm) of planar knives measured on a test stand equipped with a Hommel-Tester T8000 stylus profilometer from Hommelwerke GmbH (Villingen-Schwenningen, Germany).

Table 3 provides a collection of values of selected surface texture parameters calculated from recorded microtopographies using TalyMap Silver 4.1.2 (Digital Surf, Besançon, France) software. The collection of surface texture parameters included eight parameters from five basic groups:amplitude (arithmetic mean deviation of the surface *Sa*, total height of the surface *St*);area and volume (mean void volume ratio *Smvr*);spatial (density of summits of the surface *Sds*, texture aspect ratio of the surface *Str*);hybrid (root-mean-square slope of the surface *Sdq*, developed interfacial area ratio *Sdr*);functional (bearing index *Sbi*).

The surface texture parameters are then presented in the form of graphs (Figure 7, Figure 8, Figure 9, Figure 10, Figure 11, Figure 12, Figure 13 and Figure 14) of the individual values for the twelve blades selected for measurement, as well as a summary of the average values determined by the cooling and lubrication methods of the grinding zone used in shaping the blades.

A comparison of the determined values of the surface texture amplitude parameters shown in Figure 7 (arithmetic mean deviation of the surface *Sa*) and 8 (total height of the surface *St*) indicates that the relatively highest values were obtained under grinding conditions using the hybrid method of delivery coolants into the machining zone (MQL + CAG).

The average values of the described parameters for this method were about 30% higher compared to the other three analyzed variants (WET, MQL and CAG). On the other hand, the smallest values of surface texture amplitude parameters were registered on the surfaces of blades ground under cooling conditions using only a CAG nozzle delivering compressed cooled air. Observations of these blades indicate a significantly higher processing temperature, resulting in the formation of grinding burns and significant revind of the blade. An additional effect of the insufficient realization of the cooling function in the case of the CAG nozzle was the smoothing and blunting of the active abrasive grains, which formed a ground surface with the relatively lowest irregularities.

The analogous nature of the changes in values was also registered for other surface texture parameters: the mean void volume ratio *Smvr* (Figure 9), the root-mean-square slope of the surface *Sdq* (Figure 12), the developed interfacial area ratio *Sdr* (Figure 13) and the value of the bearing index *Sbi* (Figure 14). In the case of the texture aspect ratio of the surface *Str* (Figure 11), the highest value was obtained when the blades were ground under conditions of minimized lubrication with air-oil aerosol fed by an MQL nozzle. On the other hand, an analysis of the changes in the density of summits of the surface *Sds* presented in Figure 10 showed that the use of the MQL method in the grinding process resulted in a reduction of *Sds* values by about 12%, and for the hybrid method (MQL + CAG) by about 23% with respect to the maximum average value obtained for the flooding method (*Sds* = 6154 mm^−2^).

The results obtained from the analysis of selected surface texture parameters showed that changing the cooling and lubrication conditions of the machining zone in the grinding process of planar knife blades significantly affects the height of the shaped summits (amplitude parameters *Sa* and *St*), as well as the associated volumetric (parameter *Smvr*), spatial (parameters *Sds* and *Str*), hybrid (parameters *Sdq* and *Sdr*) and functional (parameter *Sbi*) features. The results of the measurements lead to the conclusion that the use of the WET flooding method and MQL allowed the shaping of blade surfaces with similar geometric structures. The use of only compressed-air cooling with a CAG nozzle resulted in a relative reduction in the height of summits *Sa* and *St*, the texture aspect ratio of the surface *Str*, the root-mean-square slope of the surface *Sdq* as well as the bearing index *Sbi* and not significantly affect the mean void volume ratio *Smvr* and the density of summits of the surface *Sds*. The greatest impact of this method was noted in the case of the value of developed interfacial area ratio *Sdr*, which was more than half the size of the WET flooding method. The observed differences in surface texture for the different cooling methods are due to differences in the tribological conditions of the phenomena of initiation of chip formation, separation of material particles from the workpiece surface and furrowing. The effective supply of lubricant to the machining zone in the flooding and MQL method reduces the friction between the vertices of active abrasive grains and workpieces, favorably reducing the flux of heat generated while hindering the initiation of chip formation by abrasive grains. This leads to an increase in the proportion of furrowing and the formation of chips with larger cross-sections under such conditions. The effectiveness of receiving the heat generated in the grinding zone, which differs from one method to another, significantly affects the intensity of the phenomena associated with the wear of abrasive grains (dulling, plasticization), resulting in a decrease in the texture aspect ratio and the number of vertices per area unit.

Accordingly, the use of the hybrid MQL + CAG method in the grinding process resulted in an increase in the height of summits (*Sa* and *St*), the mean void volume ratio *Smvr*, the root-mean-square slope of the surface *Sdq*, the developed interfacial area ratio *Sdr* and the bearing index *Sbi*, while reducing the density of summits of the surface *Sds* compared to average values of the surface texture parameters obtained for the flooding WET method. No clear effect of cooling and lubrication of the machining zone by the hybrid method on changes in the texture aspect ratio of the surface *Str* was noted.

Summarizing this part of the analysis, it can be concluded that by selecting the cooling and lubrication conditions of the grinding zone, the functional characteristics of the surface can be significantly influenced, which should also potentially affect the intensity of wear phenomena and ultimately, the life of blades in the process of skinning flat fish.

At this point, it is worth mentioning that the predominant phenomena determining the end of the service life of the technical blades tested in the process of skinning flat fish are chipping and edge deformation. They are caused by the impact of hard inclusions and impurities anchored in the skin of processed fish. Due to living conditions, the skin of flat fish is often reinforced with sand grains and shell particles, which are many times harder than the animal tissue being cut and are the main reason for the limited durability of planar blades under industrial production conditions.

Changes in the functional characteristics of the surface after grinding with the selected cooling and lubrication methods were also demonstrated by measurements of the cutting force *F*, the results of which are shown in Figure 5. Analysis of the surface texture parameters in Table 3 and Figure 7, Figure 8, Figure 9, Figure 10, Figure 11, Figure 12, Figure 13 and Figure 14 provided details to relate the surface characteristics of the blade to the cutting force *F* values. The surface texture of the blade surface has a decisive influence (along with the vertex angle) on the force required to decohere the material during the cutting process. It determines the magnitude of the frictional force between the surface of the blade and the material to be cut as the blade plunges into the sample. In addition, the texture properties of the blade’s side surfaces determine the geometry and structure of its cutting edge resulting from the process of shaping them. For these reasons, it should be expected that surfaces with a less developed surface and characterized by smaller values of amplitude parameters allow the formation of a more aligned blade edge with a smaller rounding radius. The results presented in this study confirm this conclusion, while it should also be emphasized that the final machining result is also strongly influenced by the method of removing the material rewind after the grinding process.

Figure 15, Figure 16, Figure 17 and Figure 18 show selected results of microscopic observations of the surfaces of planar knife blades shaped using four methods of delivery coolants to the machining zone (WET—Figure 15, MQL—Figure 16, CAG—Figure 17 and MQL + CAG—Figure 18).

Analysis of the images in Figure 15, Figure 16, Figure 17 and Figure 18 indicates very similar morphology of the surface of the blade shaped under conditions using the flooding method (WET), MQL and the hybrid method (MQL + CAG). This is confirmed by the similar force values F measured when cutting test specimens with such ground knives (Figure 5).

In the set of microscopic images analyzed, the views of the surfaces of blades shaped under conditions of compressed cooled air delivery stand out clearly (Figure 17). These blades were characterized by a very large rewinding, which was not fully removed in the procedure of completing the shaping of the blades. Microscopic images clearly show unremoved fragments of chips adhered to the blade and numerous chippings.

This testifies to unfavorable thermal conditions in the grinding zone, causing plasticization of the shaped blade. These are due to insufficient cooling, and the lack of lubricating function in the grinding process carried out under conditions of using only the CAG nozzle. As a result, the average cutting force with these blades was almost double that of other planar blades ground under flooding cooling conditions, the MQL method and the hybrid method combining the use of the MQL nozzle and the CAG nozzle (Figure 5). These observations are also confirmed by the results of the analysis of the values of selected surface texture parameters shown in Table 3 and Figure 7, Figure 8, Figure 9, Figure 10, Figure 11, Figure 12, Figure 13 and Figure 14.

Summarizing the obtained results of the analysis, it can be stated that the most favorable (smallest) values of cutting force *F* (Figure 4) and grinding power gain *ΔP* (Figure 5) were recorded for the hybrid method (MQL + CAG). However, analysis of the surface texture showed more favorable stereometric characteristics of the blades ground under lubrication conditions of the MQL method (Figure 7, Figure 8, Figure 9, Figure 10, Figure 11, Figure 12, Figure 13 and Figure 14). The use of the hybrid method resulted in obtaining a blade surface with relatively largest irregularities described by amplitude parameters (*Sa* and *St*—Figure 7 and Figure 8). On the other hand, when the MQL method was used during blade shaping, the values of the surface texture parameters were, in most cases, similar to those obtained using the flooding method (WET)—which is the reference in the applied test methodology.

## 4. Conclusions

The experimental results obtained made it possible to formulate several specific conclusions, which are given below.

Relatively small values of the grinding power gain *ΔP* were measured when the grinding process was carried out under MQL conditions (*ΔP* = 15.20 W) and using the hybrid method MQL + CAG (*ΔP* = 14.40 W). In the case of the WET and CAG methods, the *ΔP* values were about 50–60% higher, respectively *ΔP* = 22.67 W and *ΔP* = 23.20 W. The application of the MQL method and the hybrid MQL + CAG method provided a very good realization of the lubricating function so that the share of friction of dulled cutting vertices against the workpiece surface is reduced, which manifests itself in the reduction of the grinding force and the correlated grinding power.Beneficial cutting properties (manifested by relatively small *F* values) were obtained by using flooding cooling WET (*F* = 11.81 N), MQL (*F* = 15.25 N) and the hybrid method MQL + CAG (*F* = 13.69 N) in the grinding process. In the case of grinding under cooled compressed air delivery conditions, the average cutting force was as much as 91.6% higher (*F* = 22.63 N) compared to the result obtained for the most favorable flooding method, demonstrating the insufficient quality of the blade shaped under such conditions.The results of the analysis of selected parameters of the surface texture of the planar blade led to the conclusion that the application of the WET flooding method and MQL made it possible to shape the surfaces of the blade with a similar geometric structure.Analysis of microscopic images showed a very similar condition of the surfaces of blades shaped under conditions of application of the WET, MQL and MQL + CAG methods. Against this background, the views of the surfaces of blades shaped under the conditions of delivering compressed cooled air with a CAG nozzle stood out clearly, which were characterized by a very large rewind and visible unremoved chips adhered to the blade, as well as numerous chippings of the blade. These features confirm previous conclusions about unfavorable thermal conditions in the grinding zone under CAG cooling conditions causing plasticization of the shaped blade.By selecting the cooling and lubrication conditions of the grinding zone, the functional characteristics of the blade surface can be significantly influenced, which affects the measured values of the cutting force and should potentially also affect the intensity of wear phenomena and, ultimately, the life of the blades in the process of skinning flat fish.A comprehensive comparison of test results on grinding power gain *ΔP*, cutting force *F*, and surface texture suggest that the most favorable sharpening results were obtained using an environmentally friendly MQL method of cooling and lubricating the grinding zone.

## Figures and Tables

**Figure 1 materials-15-07842-f001:**
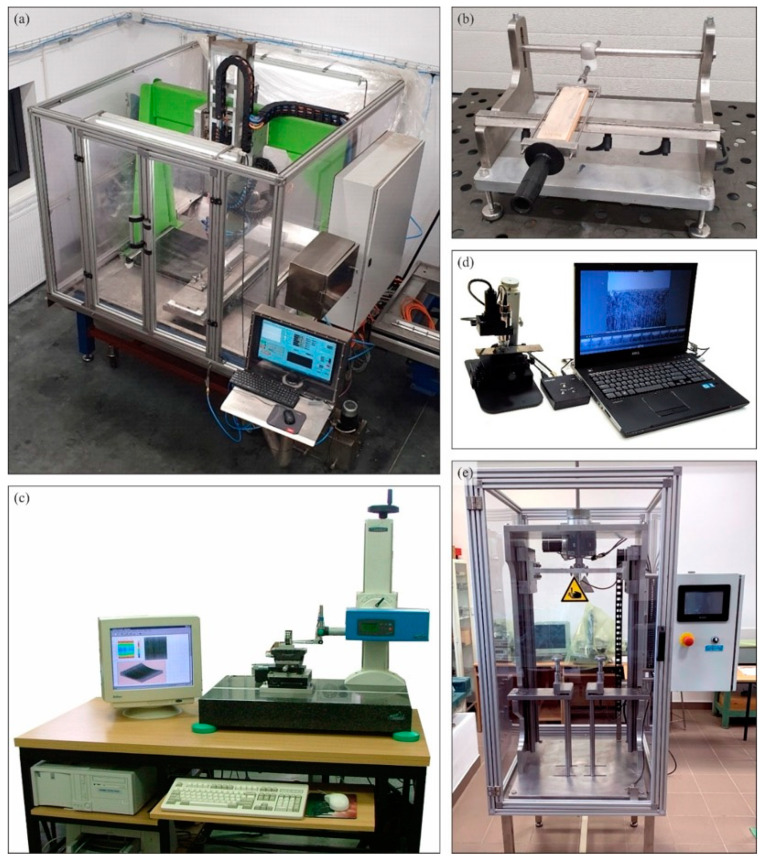
Test and measurement stands used in the described research: (**a**) five-axis CNC special grinder for sharpening technical blades; (**b**) special device for removing rewind from the blade after grinding; (**c**) position for contact measurements of surface texture equipped with a stylus profilometer Hommel-Tester T8000 by Hommelwerke GmbH, Villingen-Schwenningen, Germany [38]; (**d**) digital measuring microscope Dino-Lite Edge AM7515MT8A, AnMo Electronics Corporation, New Taipei City, Taiwan, China [39,40,41]; (**e**) special test stand for cutting force measurement.

**Figure 2 materials-15-07842-f002:**
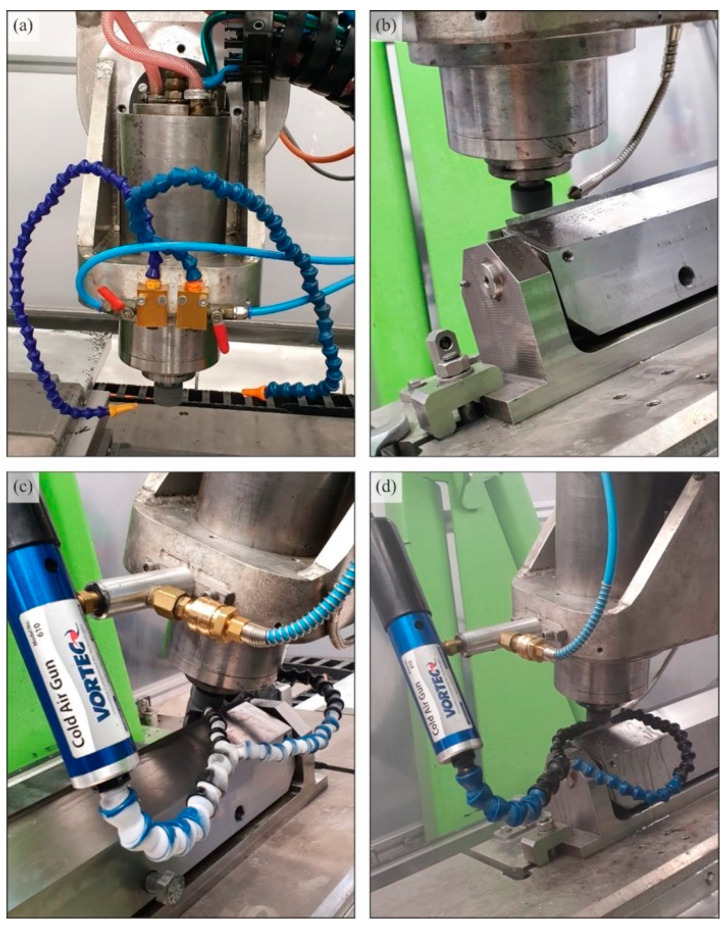
View of the grinding zone in a configuration that allows the use of (**a**) flooding method (WET); (**b**) minimum quantity lubrication (MQL); (**c**) cooling with cooled compressed air (CAG); (**d**) hybrid method involving the simultaneous use of MQL and CAG nozzle (MQL + CAG).

**Figure 3 materials-15-07842-f003:**
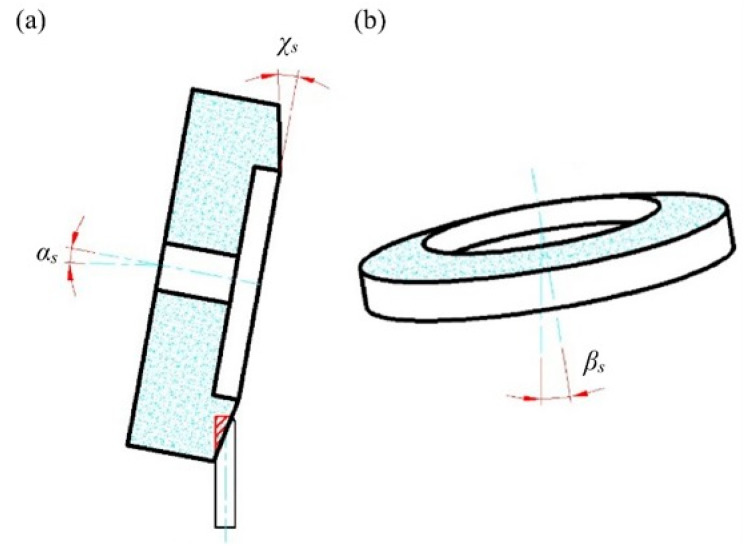
Diagram of the angular positioning of the grinding wheel in three planes with respect to the planar blade: (**a**) side view; (**b**) front view (angle values used in the described tests: *α_s_* = 85°, *β_s_* = 5°, *χ_s_* = 20°).

**Figure 4 materials-15-07842-f004:**
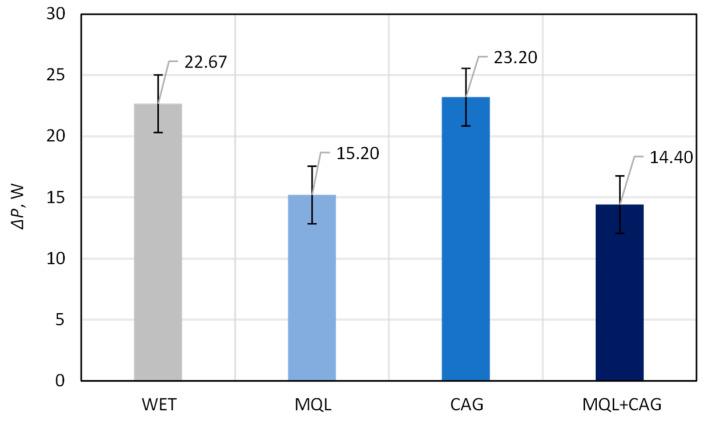
The average values of the grinding power gain *ΔP* determined during grinding tests carried out with 5 repetitions using four methods of delivering coolants to the machining zone (flooding method WET, MQL, cooling using cooled compressed air generated by CAG nozzle and the hybrid method MQL + CAG) (error bars represent the standard error equal to the standard deviation *σ* divided by the square root of the total number of samples).

**Figure 5 materials-15-07842-f005:**
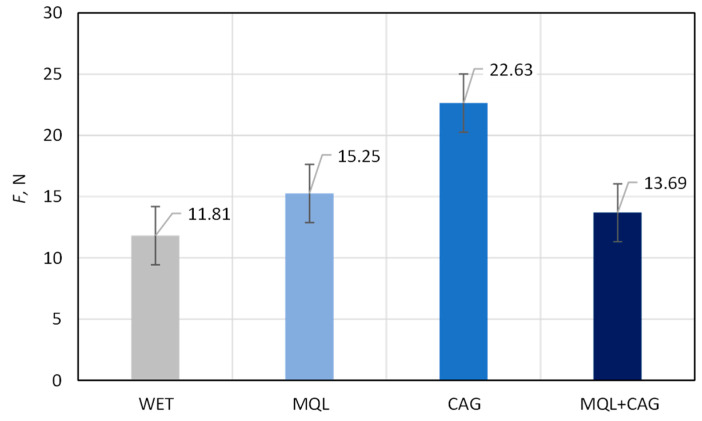
Average values (from 5 repetitions) of cutting force *F* determined from measurements of blades shaped using four methods of supplying coolants to the grinding zone: flooding method WET, MQL, cooling using cooled compressed air generated by CAG nozzle and the hybrid method MQL + CAG (error bars represent the standard error equal to the standard deviation *σ* divided by the square root of the total number of samples).

**Figure 6 materials-15-07842-f006:**
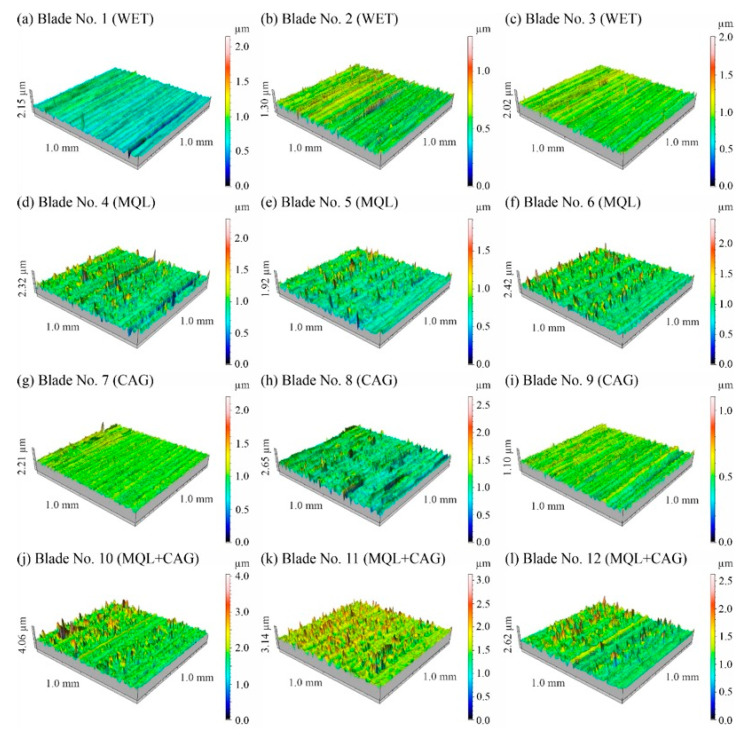
View of surface microtopography (1.0 × 1.0 mm) of planar knives measured on a test stand equipped with a Hommel-Tester T8000 stylus profilometer from Hommelwerke GmbH (Villingen-Schwenningen, Germany): (**a**–**c**) blades No. 1–3 ground under WET method conditions; (**d**–**f**) blades No. 4–6 ground under MQL method conditions; (**g**–**i**) blades No. 7–9 ground under CAG method conditions; (**j**–**l**) blades No. 10–12 ground under MQL + CAG method conditions.

**Figure 7 materials-15-07842-f007:**
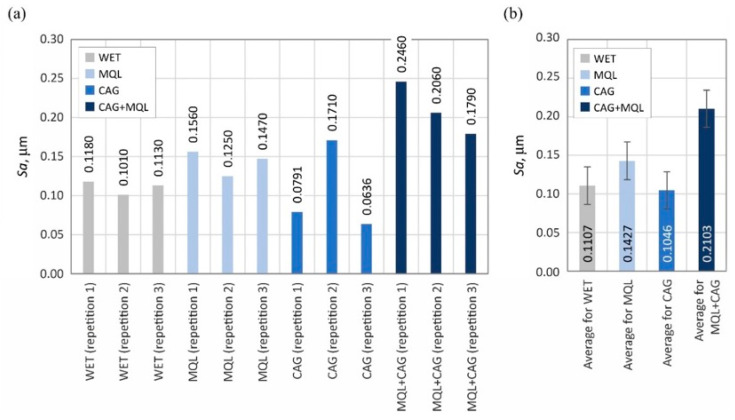
Chart of changes in the arithmetic mean deviation of the surface *Sa*: (**a**) results for the twelve blades selected for the surface texture measurements; (**b**) mean values of the parameters divided by cooling and lubrication methods of the grinding zone used during the shaping of the blades (error bars represent the standard error equal to the standard deviation *σ* divided by the square root of the total number of samples).

**Figure 8 materials-15-07842-f008:**
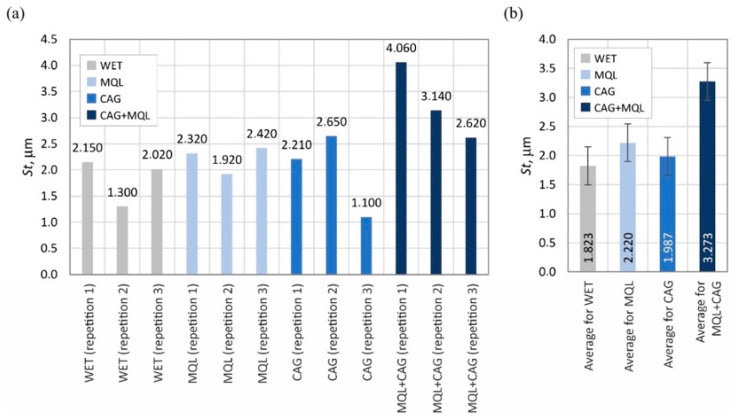
Chart of changes in the total height of the surface *St*: (**a**) results for the twelve blades selected for the surface texture measurements; (**b**) mean values of the parameters divided by cooling and lubrication methods of the grinding zone used during the shaping of the blades (error bars represent the standard error equal to the standard deviation *σ* divided by the square root of the total number of samples).

**Figure 9 materials-15-07842-f009:**
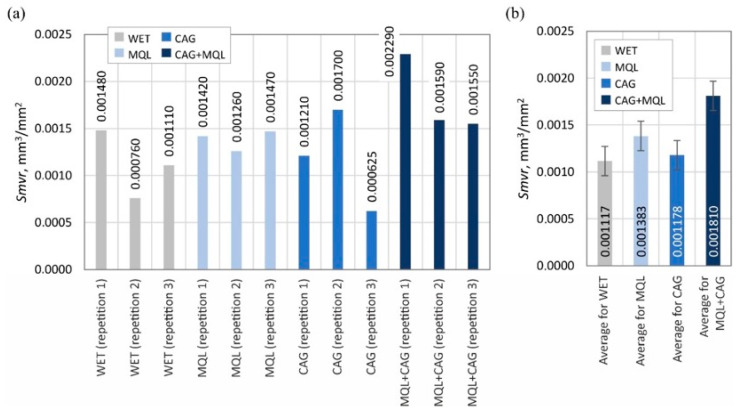
Chart of changes in the mean void volume ratio *Smvr*: (**a**) results for the twelve blades selected for the surface texture measurements; (**b**) mean values of the parameters divided by cooling and lubrication methods of the grinding zone used during the shaping of the blades (error bars represent the standard error equal to the standard deviation *σ* divided by the square root of the total number of samples).

**Figure 10 materials-15-07842-f010:**
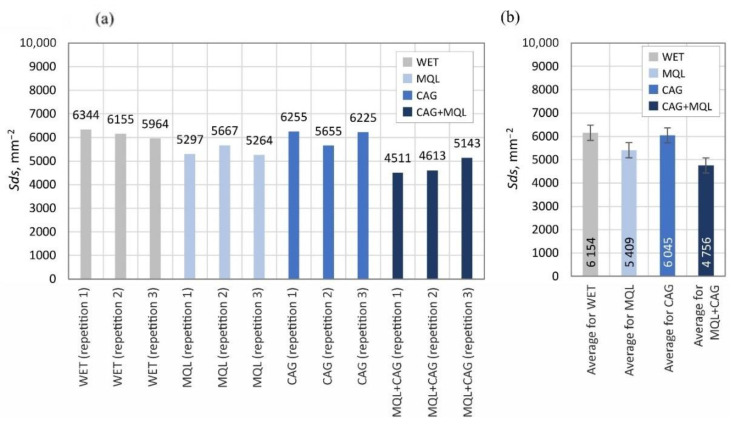
Chart of changes in the density of summits of the surface *Sds*: (**a**) results for the twelve blades selected for the surface texture measurements; (**b**) mean values of the parameters divided by cooling and lubrication methods of the grinding zone used during the shaping of the blades (error bars represent the standard error equal to the standard deviation *σ* divided by the square root of the total number of samples).

**Figure 11 materials-15-07842-f011:**
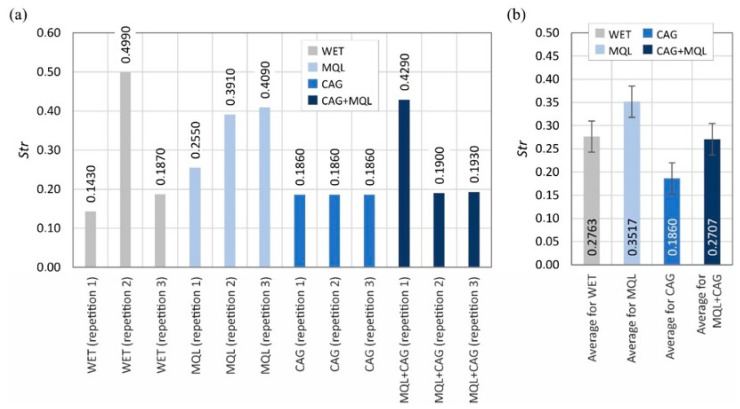
Chart of changes in the texture aspect ratio of the surface *Str*: (**a**) results for the twelve blades selected for the surface texture measurements; (**b**) mean values of the parameters divided by cooling and lubrication methods of the grinding zone used during the shaping of the blades (error bars represent the standard error equal to the standard deviation *σ* divided by the square root of the total number of samples).

**Figure 12 materials-15-07842-f012:**
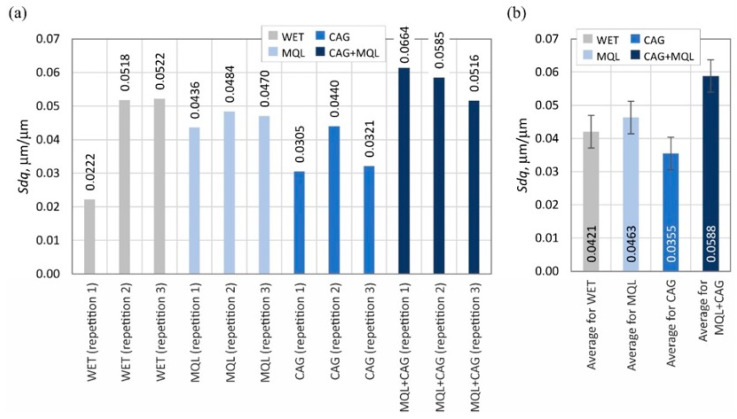
Chart of changes in the root-mean-square slope of the surface *Sdq*: (**a**) results for the twelve blades selected for the surface texture measurements; (**b**) mean values of the parameters divided by cooling and lubrication methods of the grinding zone used during the shaping of the blades (error bars represent the standard error equal to the standard deviation *σ* divided by the square root of the total number of samples).

**Figure 13 materials-15-07842-f013:**
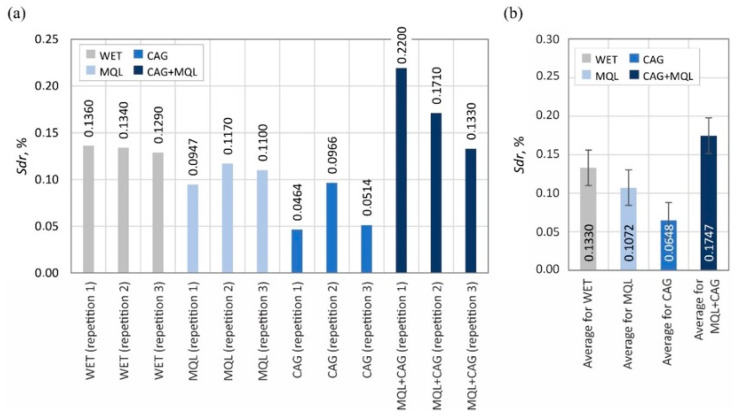
Chart of changes in the developed interfacial area ratio *Sdr*: (**a**) results for the twelve blades selected for the surface texture measurements; (**b**) mean values of the parameters divided by cooling and lubrication methods of the grinding zone used during the shaping of the blades (error bars represent the standard error equal to the standard deviation *σ* divided by the square root of the total number of samples).

**Figure 14 materials-15-07842-f014:**
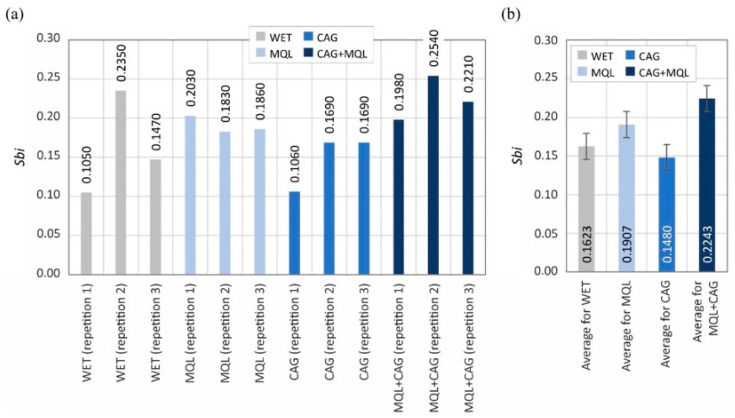
Chart of changes in the bearing index *Sbi*: (**a**) results for the twelve blades selected for the surface texture measurements; (**b**) mean values of the parameters divided by cooling and lubrication methods of the grinding zone used during the shaping of the blades (error bars represent the standard error equal to the standard deviation *σ* divided by the square root of the total number of samples).

**Figure 15 materials-15-07842-f015:**
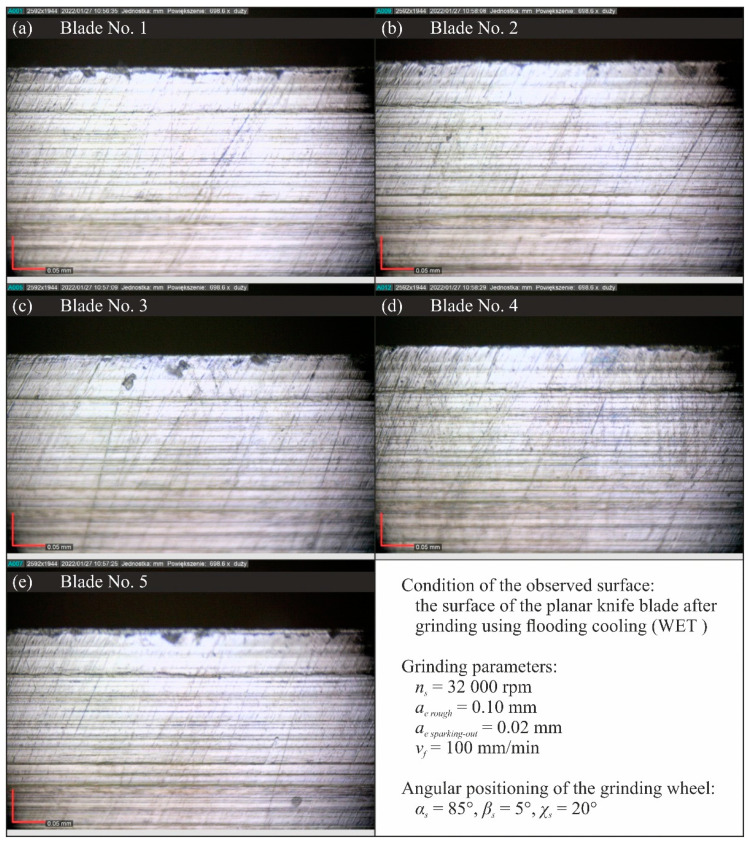
Microscopic views of the blade surfaces of five planar knives shaped using flooding (WET) cooling recorded with a Dino-Lite Edge AM7515MT8A digital measuring microscope from AnMo Electronics Corp. (New Taipei City, Taiwan, China) at a magnification of approximately 700×: (**a**) blade No. 1; (**b**) blade No. 2; (**c**) blade No. 3; (**d**) blade No. 4; (**e**) blade No. 5.

**Figure 16 materials-15-07842-f016:**
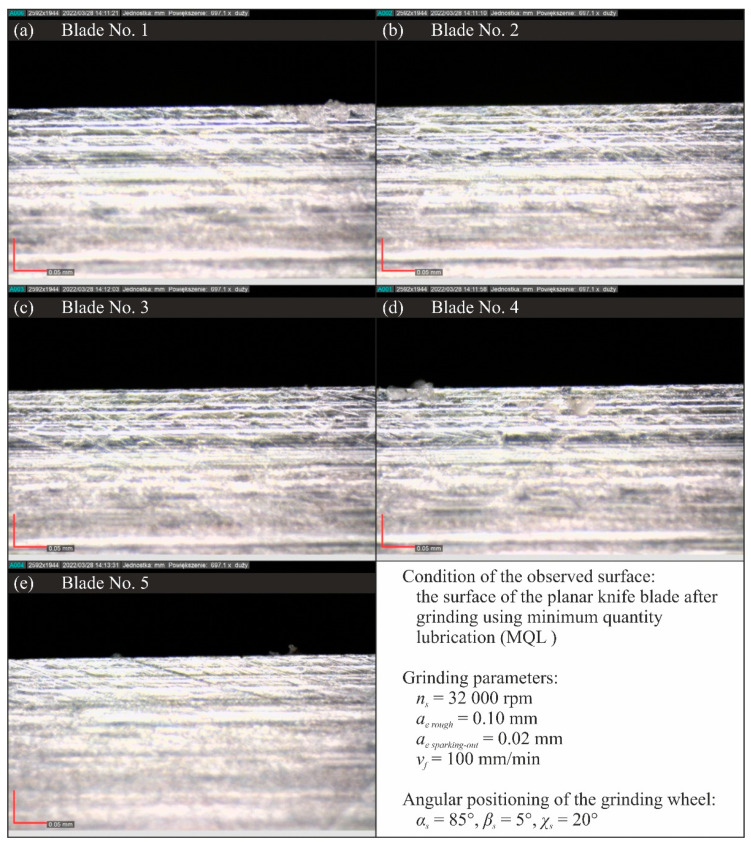
Microscopic views of the blade surfaces of five planar knives shaped using the MQL method recorded with a Dino-Lite Edge AM7515MT8A digital measuring microscope from AnMo Electronics Corp. (New Taipei City, Taiwan, China) at a magnification of approximately 700×: (**a**) blade No. 1; (**b**) blade No. 2; (**c**) blade No. 3; (**d**) blade No. 4; (**e**) blade No. 5.

**Figure 17 materials-15-07842-f017:**
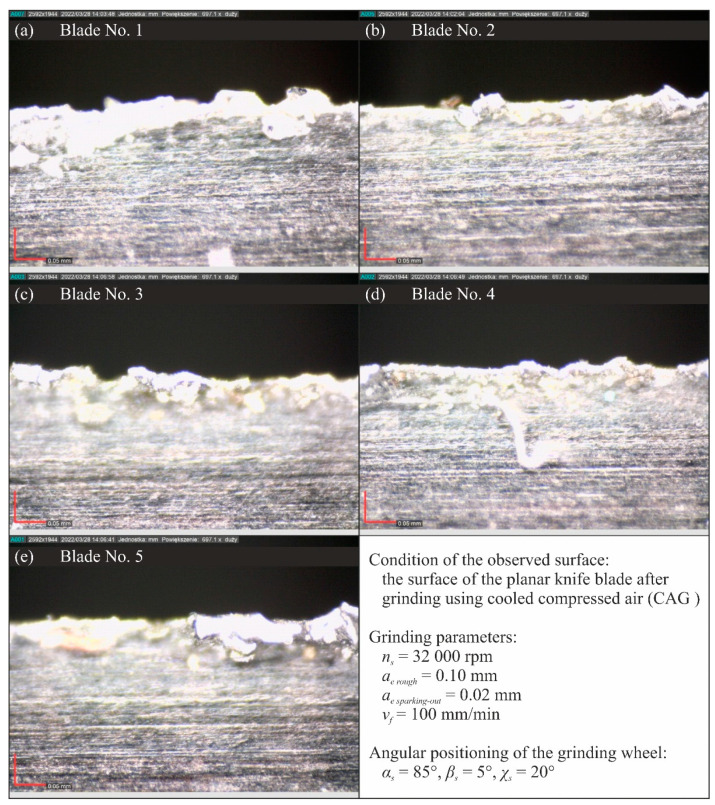
Microscopic views of the blade surfaces of five planar knives shaped using cooled compressed air (CAG) recorded with a Dino-Lite Edge AM7515MT8A digital measuring microscope from AnMo Electronics Corp. (New Taipei City, Taiwan, China) at a magnification of approximately 700×: (**a**) blade No. 1; (**b**) blade No. 2; (**c**) blade No. 3; (**d**) blade No. 4; (**e**) blade No. 5.

**Figure 18 materials-15-07842-f018:**
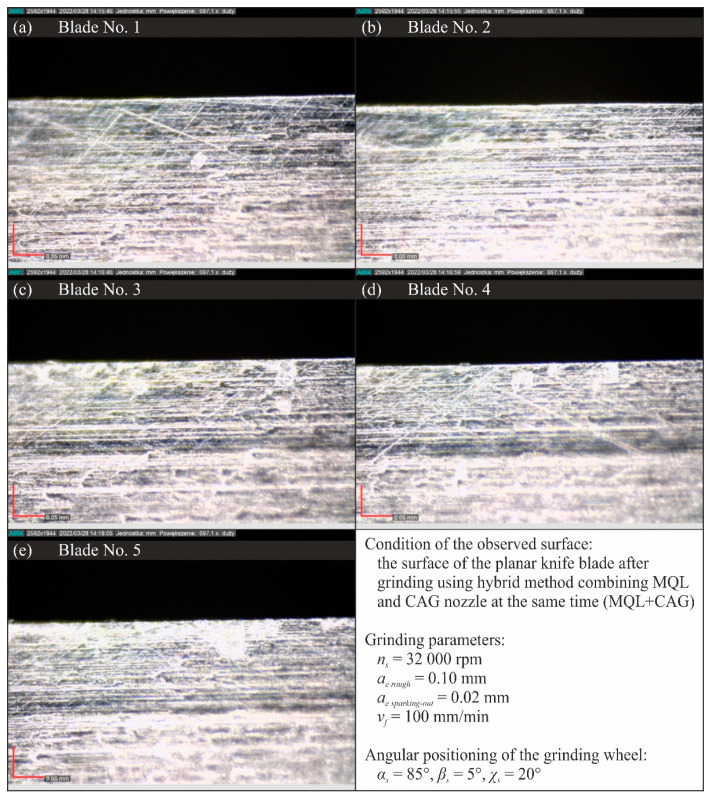
Microscopic views of the blade surfaces of five planar knives shaped using a hybrid method combining MQL and CAG nozzle at the same time (MQL + CAG) were recorded with a Dino-Lite Edge AM7515MT8A digital measuring microscope from AnMo Electronics Corp. (New Taipei City, Taiwan, China) at a magnification of approximately 700×: (**a**) blade No. 1; (**b**) blade No. 2; (**c**) blade No. 3; (**d**) blade No. 4; (**e**) blade No. 5.

**Table 1 materials-15-07842-t001:** Designation and main technical parameters of the grinding wheel [42].

Designation	Dimensions					Composition			
	*D*, mm	*P*, mm	*T*, mm	*H*, mm	*F*, mm	Grain Size according to FEPA, µm	Grain Concentration, carat/cm^3^	Degree of Hardness	Bond
5A1 35 × 25 × 10/22 × 15 B181 V240 SV	35	22	25	10	15	180/150	4.18	Medium	Vitrified
**Diagram of grinding wheel construction with designations of basic dimensions**	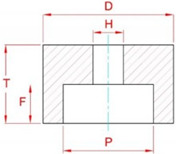								

**Table 2 materials-15-07842-t002:** Specification of parameters and conditions of experimental studies.

**Process**	Rectilinear grinding of flat surfaces	
**Test stand**	Specialized five-axis CNC grinding machine for shaping knife blades with low rigidity	
**Workpiece**	Planar knives designed for the process of skinning flat fish made of high-carbon martensitic X39Cr13 stainless steel (Kuno Wasser GmbH, Solingen, Germany for Steen F.P.M. International, Kalmthout, Belgium)	
**Grinding wheel**	5A1 35 × 25 × 10/22 × 15 B181 V240 SV (INTER-DIAMENT, Grodzisk Mazowiecki, Poland)	
**Dressing parameters of the grinding wheel**	Dresser: M1039/D 1.00 ct (Dialeks, Pruszków, Poland) Rotational speed of grinding wheel in dressing: *n_sd_* = 32,000 rpm Feed rate during dressing: *v_fd_* = 0.00165 m/s Dressing allowance: *a_d_* = 0.03 mm Number of dressing passes: *i_d_* = 2	
**Grinding parameters**	**Constant input quantities**	**Variable input quantities**
	Rotational speed of grinding wheel: *n_s_* = 32,000 rpm Allowance for rough passage: *a_e rough_* = 0.10 mm Longitudinal feed velocity of the grinding wheel: *v_f_* = 100 mm/min Number of roughing passes for both phases of the blade: 1 Allowance for sparking-out passage: *a_e sparking-out_* = 0.02 mm Number of sparking-out passes for both phases of the blade: 1 Direction of rotation of the grinding wheel: right—in the direction of the axis of symmetry of the knife (to the blade) Grinding kinematics: grinding with the conical surface of the grinding wheel Angular positioning of the grinding wheel in the grinding process: *α_s_* = 85°, *β_s_* = 5°, *χ_s_* = 20°	Conditions for delivering cooling and lubricating agents to the grinding zone: flooding method (WET), minimum quantity lubrication MQL, cooling using cooled compressed air generated by CAG nozzle and the hybrid method combining MQL and CAG nozzle (MQL + CAG)
**Cooling and lubrication conditions of the grinding zone**	Flooding (WET) cooling using a low-pressure circular nozzle with expenditure *Q* = 1.75 dm^3^/min. Coolant: 5% water-oil emulsion of Cimtech^®^ M26 oil by CIMCOOL^®^ Fluid Technology forming part of Milacron LLC (Cincinnati, OH, USA)	
	Minimum quantity lubrication using ZMIN-MS nozzle by Sommer-Technik GmbH (Straubenhardt, Germany) with expenditure *Q_MQL_* = 1100 mL/h. Air pressure feeding the nozzle: 0.8 MPa. Coolant: air-oil aerosol (Cimtech^®^ MQL oil by CIMCOOL^®^ Fluid Technology forming part of Milacron LLC, Cincinnati, OH, USA)	
	Cooling using cooled compressed air generated by a Vortec 610 CAG nozzle by ITW Vortec and Paxton Products (Cincinnati, OH, USA) with expenditure *Q_CAG_* = 49.8 dm^3^/min. Air pressure feeding the nozzle: 0.6 MPa. Coolant: cooled compressed air	

**Table 3 materials-15-07842-t003:** Collection of values of selected surface texture parameters of planar knives shaped in experimental studies determined from measurements of blade surface microtopography.

Designation of the Blade Selected for Analysis	Surface Texture Parameters							
	Amplitude		Area and Volume	Spatial		Hybrid		Functional
	*Sa*, µm	*St*, µm	*Smvr*, mm^3^/mm^2^	*Sds*, mm^−2^	*Str*	*Sdq*, µm/µm	*Sdr*, %	*Sbi*
WET (repetition 1)	0.1180	2.150	0.001480	6344	0.1430	0.0222	0.1360	0.1050
WET (repetition 2)	0.1010	1.300	0.000760	6155	0.4990	0.0518	0.1340	0.2350
WET (repetition 3)	0.1130	2.020	0.001110	5964	0.1870	0.0522	0.1290	0.1470
MQL (repetition 1)	0.1560	2.320	0.001420	5297	0.2550	0.0436	0.0947	0.2030
MQL (repetition 2)	0.1250	1.920	0.001260	5667	0.3910	0.0484	0.1170	0.1830
MQL (repetition 3)	0.1470	2.420	0.001470	5264	0.4090	0.0470	0.1100	0.1860
CAG (repetition 1)	0.0791	2.210	0.001210	6255	0.1860	0.0305	0.0464	0.1060
CAG (repetition 2)	0.1710	2.650	0.001700	5655	0.1860	0.0440	0.0966	0.1690
CAG (repetition 3)	0.0636	1.100	0.000625	6225	0.1860	0.0321	0.0514	0.1690
MQL + CAG (repetition 1)	0.2460	4.060	0.002290	4511	0.4290	0.0664	0.2200	0.1980
MQL + CAG (repetition 2)	0.206	3.140	0.001590	4613	0.1900	0.0585	0.1710	0.2540
MQL + CAG (repetition 3)	0.179	2.620	0.001550	5143	0.1930	0.0516	0.1330	0.2210

## Data Availability

Not applicable.

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
