# Peer review of "Effect of Pro-Ecological Cooling and Lubrication Methods on the Sharpening Process of Planar Blades Used in Food Processing"

_materials, 2022, doi:10.3390/ma15217842_

Round 1
Reviewer 1 Report
1. Background should be shortened in introduction
2. What is the purpose to take the hybrid method MQL+CAG into consideration?
3. Conclusions should be reorganized and concised
4. Format should be improved based on the template.
5. What is the main failure types of the employed blades?
6. What is the innovation of this manuscript?
Author Response
The Authors wish to thank all the Reviewers for their time spent on prepare review of the manuscript “Effect of pro‑ecological cooling and lubrication methods on the sharpening process of planar blades used in food processing”. All the valuable comments, suggestions and hints were very helpful in improving the readability of the text and in improving of its scientific quality. Below the detailed responses upon these subsequent comments were given. The modified or added text in the manuscript was highlighted in red.
REVIEWER 1
Reviewer comment 1:
Background should be shortened in introduction.
Authors response:
Introduction was shortened according to the reviewer’s comment.
Reviewer comment 2:
What is the purpose to take the hybrid method MQL+CAG into consideration?
Authors response:
The use of the MQL method allows effective lubrication of the grinding zone, but in many cases does not provide adequate heat dissipation due to the limited contribution of the lubricant. On the other hand, the use of cooled compressed air generated by the CAG nozzle allows cooling of the machining zone with a marginal realization of the lubricating function. Thus, it seems natural to combine these two methods (in one hybrid MQL+CAG method), whose features are complementary.
The quoted description was added in the text of the revised version of the article.
Reviewer comment 3:
Conclusions should be reorganized and concised.
Authors response:
In the revised version of the article the conclusions concerning the individual areas of analysis have been combined. Some of the conclusions were removed and as a result their number was reduced to six.
Reviewer comment 4:
Format should be improved based on the template.
Authors response:
The authors have made every effort to ensure that the submitted article fully meets the guidelines of the journal and fits the template. We would additionally like to note that the final layout of the text for publication will still be modified by the publisher.
Reviewer comment 5:
What is the main failure types of the employed blades?
Authors response:
The predominant phenomena determining the end of the service life of the technical blades tested in the process of skinning flat fish are chipping and edge deformation. They are caused by the impact of hard inclusions and impurities anchored in the skin of processed fish. Due to living conditions, the skin of flat fish is often reinforced with sand grains and shell particles, which are many times harder than the animal tissue being cut and are the main reason for the limited durability of planar blades under industrial production conditions.
The quoted description was added in the text of the revised version of the article.
Reviewer comment 6:
What is the innovation of this manuscript?
Authors response:
In the directional literature there are very few descriptions of research work on the shaping of technical blades. It seems that the development of blade shaping techniques is carried out within the framework of corporations that do not care about the popularization of knowledge only about the commercialization of the results of their research. As a result, the issue presented in the submitted article is a complete novelty and has not been described before. While can be found hundreds of works on pro-ecological methods of cooling and lubricating the machining zone, none of them deals with the process of sharpening knives. For this reason, the presented research results can be considered innovative.

Reviewer 2 Report
In this work, the authors introduce interesting experimental research, which compares different cooling and lubrication methods during planar blades sharpening process. The aim of this work is to find out an efficient approach to minimise the consumption of the cooling and lubrication expenditure, considering the environmental effect and human health. The structure of the contents in each section of this paper is clear. Key information regarding the experimental setup, test and measurement equipment, and materials is provided.
However, the authors are required to re-organise the detailed content mentioned below. In addition, the main results need to be discussed more. The conclusion should point out the preferred cooling and lubrication method at the end of the research to guide practical food processing.
1 The abstract in this paper is too long. The authors spend too many words introducing the background of the research. However, the detailed effect of each cooling and lubrication method, the key results from the research, and the benefit of the preferred method, which are significant content in the abstract are obscure.
2 In Table 1, the “diagram of grinding wheel construction with designations of basic dimensions” should be corrected. The outline of the grinding wheel in Figure 3 (a) shows an angle χs. However, it is flat in the diagram in Table 1.
3 The key information about the experimental equipment and material is presented in Table 2. However, it contains too much information. The authors should use one paragraph to describe the related information, but only put key parameters in the table.
4 The sentence between the lines 288-290 is hard to understand. It is recommended to restructure this one.
5 The acronym names should be used consistently without repeating the full names (for example, remove minimum quantity lubrication, but only use MQL instead in 318).
6 The discussion of the results between the lines 311- 325 should be refined to strengthen the main conclusion related to each method. In addition, the advantages of the well-behaved method shown from the result should be explicitly described.
7 The authors do not mention how the grinding power and cutting force are measured in the paper. In Figure1 (a) (e), the special grinder and cutting force measurement equipment are not properly shown.
8 The scale of microtopographies in Figure. 6 is not clear. The authors should make the text and scale in the subfigures visible. In addition, the experiments in each group were repeated five times. The authors should explain why they measure only three blades for each group. Since there is not too much research data in this work, the authors should implement more experiments (changing rotational speed and velocity of grinding wheel) to generate sufficient data to be convincing.
9 The authors provide different surface texture parameters (From Figure 7-14) in the “Result and Discussion” section, which show how the functional surfaces are correlated with the cooling and lubrication conditions. However, it is not clear how these parameter changes would affect the cutting force F as shown in line 377. The authors should explain this key statement.
10 In the conclusion, the statement of the “smallest values” in line 426 is not appropriate. The smallest can only refer to one thing. Similar to the statement in line 440, there are four approaches compared in this work, three of these are “most favourable” is not correct literally.
11 Finally, the authors should point out which cooling and lubrication method should be applied in shaping the planer blades for processing the skinning flat fish.

Author Response
The Authors wish to thank all the Reviewers for their time spent on prepare review of the manuscript “Effect of pro‑ecological cooling and lubrication methods on the sharpening process of planar blades used in food processing”. All the valuable comments, suggestions and hints were very helpful in improving the readability of the text and in improving of its scientific quality. Below the detailed responses upon these subsequent comments were given. The modified or added text in the manuscript was highlighted in red.
REVIEWER 2
In this work, the authors introduce interesting experimental research, which compares different cooling and lubrication methods during planar blades sharpening process. The aim of this work is to find out an efficient approach to minimise the consumption of the cooling and lubrication expenditure, considering the environmental effect and human health. The structure of the contents in each section of this paper is clear. Key information regarding the experimental setup, test and measurement equipment, and materials is provided.
However, the authors are required to re-organise the detailed content mentioned below. In addition, the main results need to be discussed more. The conclusion should point out the preferred cooling and lubrication method at the end of the research to guide practical food processing.
Reviewer comment 1:
The abstract in this paper is too long. The authors spend too many words introducing the background of the research. However, the detailed effect of each cooling and lubrication method, the key results from the research, and the benefit of the preferred method, which are significant content in the abstract are obscure.
Authors response:
In the revised version of the article, the reviewer's comments were fully considered, and the modified abstract was rewritten as follows:
“This work presents the results of an experimental study of the sharpening of planar technical blades used in the fish processing industry. Sharpening was carried out in the grinding process using several environmentally friendly methods of cooling and lubricating the machining zone (MQL method, CAG nozzle, hybrid method that is a combination of MQL and CAG methods as well as WET flooding method as reference). The purpose of the research was to determine the possibility of reducing the negative environmental impact of the sharpening process of technical blades by minimizing the expenditure of coolant. The application of the MQL method and the hybrid MQL+CAG method provided a very good realization of the lubricating function, so that the share of friction of dulled cutting vertices against the workpiece surface is reduced, which manifests itself in the reduction of the grinding force and the correlated grinding power. In the case of grinding under cooled compressed air delivery conditions, the average cutting force was as much as 91.6% higher compared to the result obtained for the most favorable flooding method, demonstrating the insufficient quality of the blade shaped under such conditions. A comprehensive comparison of test results on grinding power gain, cutting force and surface texture suggests that the most favorable sharpening results were obtained using a hybrid environmentally friendly method of cooling and lubricating the grinding zone (MQL+CAG).”
Reviewer comment 2:
In Table 1, the “diagram of grinding wheel construction with designations of basic dimensions” should be corrected. The outline of the grinding wheel in Figure 3 (a) shows an angle χs. However, it is flat in the diagram in Table 1.
Authors response:
The diagram of grinding wheel construction with designations of basic dimensions shown in Table 3 is taken directly from the manufacturer's catallogue [38] and shows the base wheel on which the conical chamfer was shaped in the dressing procedure. Information about this modification is explicitly stated in the text describing Figure 3. The posted diagram is valid for the geometric dimensions of the base tool given in the initial part of Table 3, and its modification would limit the unambiguous interpretation of this information. For this reason, the diagram in question was left unchanged.
Reviewer comment 3:
The key information about the experimental equipment and material is presented in Table 2. However, it contains too much information. The authors should use one paragraph to describe the related information, but only put key parameters in the table.
Authors response:
According to good practice regarding the description of scientific research, researchers are required to provide precise and complete methodological information that allows other researchers to reproduce the experiments described. For this reason, a common objection to the authors of research work is the omission of important information that allows verification of the experiment in other research centers. Many years of experience in the research work of the authors of the present work resulted in a serious approach to this issue and as complete identification of the key conditions of the study as possible. In our opinion, omitting or limiting this precisely stated information will negatively affect the integrity of the presented research and limit the scientific value of the text. For these reasons, it was decided to leave Table 2 in its current form despite the reviewer's comment.
Reviewer comment 4:
The sentence between the lines 288-290 is hard to understand. It is recommended to restructure this one.
Authors response:
The indicated sentence has been rewritten and simplified in the revised version of the article in accordance with the reviewer's comment.
Reviewer comment 5:
The acronym names should be used consistently without repeating the full names (for example, remove minimum quantity lubrication, but only use MQL instead in 318).
Authors response:
The revised version of the article takes this comment into account by making more than a dozen editorial modifications omitting unnecessary repetition of acronym explanations.
Reviewer comment 6:
The discussion of the results between the lines 311- 325 should be refined to strengthen the main conclusion related to each method. In addition, the advantages of the well-behaved method shown from the result should be explicitly described.
Authors response:
In the revised version of the article, the reviewer's comment was considered, and the above-mentioned description was added.
“The observed differences in surface texture for the different cooling methods are due to differences in the tribological conditions of the phenomena of initiation of chip formation, separation of material particles from the workpiece surface and furrowing. The effective supply of lubricant to the machining zone in the flooding and MQL method reduces the friction between the vertices of active abrasive grains and workpieces favorably reducing the flux of heat generated, while hindering the initiation of chip formation by abrasive grains. This leads to an increase in the proportion of furrowing and the formation of chips with larger cross sections under such conditions. The effectiveness of receiving the heat generated in the grinding zone, which differs from one method to another, significantly affects the intensity of the phenomena associated with the wear of abrasive grains (dulling, plasticization), resulting in a decrease in the texture aspect ratio and the number of vertices per area unit.”
Reviewer comment 7:
The authors do not mention how the grinding power and cutting force are measured in the paper. In Figure1 (a) (e), the special grinder and cutting force measurement equipment are not properly shown.
Authors response:
As stated in the text of the article, grinding power was measured using software controlling the operation of the 5-axis numerically controlled grinding machine used in the study. A detailed description of this stand is included in the work [37], which is entirely devoted to its design and capabilities and is available in open access. To determine the cutting force, a special test stand was designed and constructed, which in turn was characterized and tested in detail in the paper [43] (also published in open access) giving there also a detailed description of the measurement methodology and measurement samples. For this reason, a more extensive description of the grinder used and the test stand for measuring the cutting force is omitted from this text.
Reviewer comment 8:
The scale of microtopographies in Figure. 6 is not clear. The authors should make the text and scale in the subfigures visible. In addition, the experiments in each group were repeated five times. The authors should explain why they measure only three blades for each group. Since there is not too much research data in this work, the authors should implement more experiments (changing rotational speed and velocity of grinding wheel) to generate sufficient data to be convincing.
Authors response:
The legibility of the scale in Figure 6 has been corrected in the revised version of the article. Surface microtopography measurements were carried out for randomly selected representatives of each group of ground blades due to limited access to the measurement system and the high time-consumption of such measurements. This study contains a complete set of measurement data recorded during the described stage of research. In the next stage (currently underway), research is being conducted to determine the durability of the shaped blades under conditions of industrial operation. Therefore, the authors are not in a position to supplement the set of results with suggested additional analyses.
Reviewer comment 9:
The authors provide different surface texture parameters (From Figure 7-14) in the “Result and Discussion” section, which show how the functional surfaces are correlated with the cooling and lubrication conditions. However, it is not clear how these parameter changes would affect the cutting force F as shown in line 377. The authors should explain this key statement.
Authors response:
In accordance with the comment provided, the following description was added in the mentioned passage of the article, clarifying the relationship between surface texture and blade cutting force.
“The surface texture of the blade surface has a decisive influence (along with the vertex angle) on the force required to decohere the material during the cutting process. It determines the magnitude of the frictional force between the surface of the blade and the material to be cut as the blade plunges into the sample. In addition, the texture properties of the blade's side surfaces determine the geometry and structure of its cutting edge resulting from the process of shaping them. For these reasons, it should be expected that surfaces with a less developed surface and characterized by smaller values of amplitude parameters allow the formation of a more aligned blade edge with a smaller rounding radius. The results presented in this study confirm this conclusion, while it should also be emphasized that the final machining result is also strongly influenced by the method of removing the material rewind after the grinding process.”
Reviewer comment 10:
In the conclusion, the statement of the “smallest values” in line 426 is not appropriate. The smallest can only refer to one thing. Similar to the statement in line 440, there are four approaches compared in this work, three of these are “most favourable” is not correct literally.
Authors response:
Conclusions 1 and 3 have been rewritten in accordance with the reviewer's fair comment.
Reviewer comment 11:
Finally, the authors should point out which cooling and lubrication method should be applied in shaping the planer blades for processing the skinning flat fish.
Authors response:
The revised version of the article takes into account this fair comment of the reviewer by adding a new conclusion written last in the text.

Reviewer 3 Report
1. Why CAG lubrication showed worst quality? Also which parameter is discussed for this statement? Line 253-255.
2. Figure 6 scales are blurry. Also, it seems the scales are different for almost all subfigures of Figure 6. Comparison is not fair with different scales.
3. Why hybrid MQL-CAG shows lowest power requirement? Power required for compressor during MQL is considered in this?
4. More in depth discussion is required. Currently discussion is only at interface.
5. What about the difference in required cost? Is it feasible at lab or industry scales?
6. Kindly refer the given paper for in depth understanding of use of MQL in grinding.
State of art on minimum quantity lubrication in grinding process. Materials Today: Proceedings, 5(9), 19638−19647. (DOI: 10.1016/j.matpr.2018.06.326).
Author Response
The Authors wish to thank all the Reviewers for their time spent on prepare review of the manuscript “Effect of pro‑ecological cooling and lubrication methods on the sharpening process of planar blades used in food processing”. All the valuable comments, suggestions and hints were very helpful in improving the readability of the text and in improving of its scientific quality. Below the detailed responses upon these subsequent comments were given. The modified or added text in the manuscript was highlighted in red.
REVIEWER 3
Reviewer comment 1:
Why CAG lubrication showed worst quality? Also which parameter is discussed for this statement? Line 253-255.
Authors response:
The indicated passage of the text describes the results of measurements of the cutting force F. The obtained test results described in the work indicate a much higher value of this parameter (compared to other cooling methods), which is a direct result of the quality of the shaped surface under the conditions of using a CAG nozzle. As stated later in the work, this is due to the insufficient cooling function and lack of lubrication in the grinding zone, which resulted in the formation of a blade with a very uneven surface structure illustrated in the microscopic images (Fig. 17).
Reviewer comment 2:
Figure 6 scales are blurry. Also, it seems the scales are different for almost all subfigures of Figure 6. Comparison is not fair with different scales.
Authors response:
The legibility of the scale in Figure 6 has been corrected in the revised version of the article. Unfortunately, the software used allowed the generation of 3D microtopography views of the machined surface, in which the color scale is automatically matched to the ordinates of the measured surface. This is a commonly accepted way of presenting this type of data, and the authors had no possibility of correcting it at the current stage of the research.
Reviewer comment 3:
Why hybrid MQL-CAG shows lowest power requirement? Power required for compressor during MQL is considered in this?
Authors response:
In the research described, the electrical power consumption of the grinding wheel spindle was measured, which can be interpreted as grinding power. This power results from the resistance that the workpiece material puts up during the grinding process, which means it is correlated with the grinding force. Therefore, this parameter was treated as a source of information about the conditions in the machining zone, and not as an energy indicator (power consumption as such). Accordingly, the power associated with the compression of air supplying the MQL and CAG nozzle was not included in the analyses. The power values during grinding under the conditions of the MQL method and the hybrid method MQL+CAG were very similar, which can be explained by the effective formation of an oil film on the surface of the workpiece and the grinding wheel by the MQL nozzle feeding the air-oil aerosol, providing reduced friction in the grinding zone. The originally submitted version of the article includes an analysis explaining this phenomenon:
“The use of the MQL method alone and in combination with the CAG nozzle provided a very good realization of the lubricating function (in this method, an air-oil aerosol is supplied). Under conditions of good lubrication, the proportion of friction of dulled cutting vertices against the workpiece surface is reduced, which manifests itself as a reduction in grinding force and the correlated power, measured in the described studies as the electrical power of the grinding wheel spindle.”
Reviewer comment 4:
More in depth discussion is required. Currently discussion is only at interface.
Authors response:
The revised version of the article includes additional descriptions to deepen the analysis of the research results presented. These descriptions also resulted from the comments of the other reviewers.
Reviewer comment 5:
What about the difference in required cost? Is it feasible at lab or industry scales?
Authors response:
Analysis of the costs associated with changing the cooling and lubrication conditions of the machining zone was not the subject of this study, and the authors do not have precise data on this subject. The use of the MQL and CAG methods is associated with the necessity of purchasing special nozzles and oil mist separators. The cost of purchasing MQL and CAG nozzles is relatively small (less than $1,000 each). It is also assumed that most modern industrial plants have a central compressed air system that can supply MQL and CAG nozzles. As a result, the costs of implementing pro-environmental methods of cooling and lubricating the grinding zone are relatively small and will be offset by the savings resulting from a dramatic reduction in expenditures related to the purchase and disposal of coolant. Solutions of this type are a frequent topic of scientific analyzes and are implemented in industrial practice, which is confirmed by the increasing offer of commercially available MQL and CAG systems.
Reviewer comment 6:
Kindly refer the given paper for in depth understanding of use of MQL in grinding.
State of art on minimum quantity lubrication in grinding process. Materials Today: Proceedings, 5(9), 19638−19647. (DOI: 10.1016/j.matpr.2018.06.326).
Authors response:
Thank you for pointing out a valuable source about the MQL method, which was cited in the modified version of the article.

Round 2
Reviewer 1 Report
Accept
Author Response
The Authors wish to thank all the Reviewers for their time spent on prepare review of the manuscript “Effect of pro‑ecological cooling and lubrication methods on the sharpening process of planar blades used in food processing”. All the valuable comments, suggestions and hints were very helpful in improving the readability of the text and in improving of its scientific quality. Below the detailed responses upon these subsequent comments were given. The modified or added text in the manuscript was highlighted in red.
REVIEWER 2
The authors have made changes to the manuscript based on the comments. The quality of the manuscript has improved as a result. However, there are two points to be addressed before the manuscript is qualified for publication.
Reviewer comment 1:
In line 116, the authors mentioned that the purpose of this research aims to find the minimum expenditure of coolant. However, the measurement of the coolant consumed in the experiment is missed in this research.
Authors response:
The purpose of the study was not to seek a value for the minimum coolant expenditure, but to determine whether it is possible to replace the traditional flooding method with one of the pro-ecological methods of cooling and lubricating the machining zone (characterized by minimizing the expenditure of cooling-lubrication agents). To clarify this point, the purpose of the work has been rewritten in a revised version of the article.
Reviewer comment 2:
The authors should explain why the MQL is the most favorable approach. According to the results in Figure 4 and 5, the grinding power gain and the cutting force using MQL + CAG is better than that of using MQL. The authors are required to explain this.
Authors response:
As described extensively in the work, the most favorable processing results were obtained when using the MQL method and the hybrid method (MQL+CAG). Indeed, the most favorable (smallest) values of cutting force F (Fig. 4) and grinding power gain ΔP (Fig. 5) were recorded for the hybrid method. However, analysis of the surface texture showed more favorable stereometric characteristics of the blades ground under lubrication conditions of the MQL method (Figs. 7-14). The use of the hybrid method resulted in obtaining a blade surface with relatively largest irregularities described by amplitude parameters (Sa and St – Figs. 7-8). On the other hand, when the MQL method was used during blade shaping, the values of the surface texture parameters were in most cases similar to those obtained using the flooding method (WET) – which is the reference in the applied test methodology. The revised version of the article adds a brief commentary clarifying this issue.
Reviewer comment 3:
The content of the manuscript still needs more refinement. Some minor mistakes can be found, such as:
- There is no period in line 52 after the reference [4].
- In line 97, what is the "among other factors" referring to.
- The description of the equipment between line 180 and 185 are repetitive with those in Figure 1.
Authors response:
Thank you for pointing out the editorial errors. The period was added in line 52. The phrase “among others” was removed from the sentence written in line 97 and the text included in lines 180-185 was rewritten removing repetitive information.

Reviewer 2 Report
The authors have made changes to the manuscript based on the comments. The quality of the manuscript has improved as a result.
However, there are two points to be addressed before the manuscript is qualified for publication.
1. In line 116, the authors mentioned that the purpose of this research aims to find the minimum expenditure of coolant. However, the measurement of the coolant consumed in the experiment is missed in this research.
2. The authors should explain why the MQL is the most favorable approach. According to the results in Figure 4 and 5, the grinding power gain and the cutting force using MQL + CAG is better than that of using MQL. The authors are required to explain this.
The content of the manuscript still needs more refinement. Some minor mistakes can be found, such as:
1. There is no period in line 52 after the reference [4].
2. In line 97, what is the "among other factors" referring to.
3. The description of the equipment between line 180 and 185 are repetitive with those in Figure 1.
The authors are required to improve the manuscript based on the comments mentioned above.

Author Response

(The authors gave the same response as above.)
